# Prevalence of Text Neck Between Different Female Students at Taibah University, Saudi Arabia; Cross Section Design

**DOI:** 10.3390/healthcare13060651

**Published:** 2025-03-17

**Authors:** Abdulrhman Mashabi, Walaa Ragab, Ghaida Alrayes, Joud Amoudi, Soundos Tokhta, Ghaida Bashawri, Ruqayyah Alqaznili, Jana Ghaith, Lama Almadani, Marwan M. A. Aljohani, Abdullah Al-Shenqiti

**Affiliations:** Department of Physical Therapy, College of Medical Rehabilitation Sciences, Taibah University, Al-Madinah Al-Munawarah 42353, Saudi Arabia; wragab@taibahu.edu.sa (W.R.); alrayesghaidaa@gmail.com (G.A.); amoudijoud2@gmail.com (J.A.); soundostokhta@gmail.com (S.T.); g.a.bashawri@gmail.com (G.B.); pt.ruqayyahq@gmail.com (R.A.); ptjana22@gmail.com (J.G.); lamaa.almadani@gmail.com (L.A.); mmjohani@taibahu.edu.sa (M.M.A.A.); ashengeti@taibahu.edu.sa (A.A.-S.)

**Keywords:** text neck, tragus to wall distance, CVA, PostureCo

## Abstract

**Background**: Text neck is a common postural malalignment that significantly affects daily activities. Reliable and user-friendly assessment tools are essential for physiotherapists, and evaluating text neck prevalence in both sitting and standing positions is crucial for effective rehabilitation. This study aimed to compare text neck prevalence between students in practical and non-practical disciplines using multiple assessment methods. **Methodology**: A total of 82 female students (18–24 years) from Taibah University participated. The practical group included students from the Colleges of Medical Rehabilitation Sciences and Applied Medical Sciences, who engage in hands-on learning. The non-practical group consisted of Business Administration students, whose coursework is primarily theoretical. Text neck was assessed using observational methods, tragus to wall distance (TWD), craniovertebral angle (CVA) via a smartphone application, and PostureCo in both sitting and standing positions. **Results**: Observational assessment showed text neck prevalence of 95.5% in the non-practical group and 70% in the practical group while sitting, and 91% versus 80% while standing. Based on TWD, prevalence was 48% in non-practical students and 32% in practical students. Using Physiomaster, prevalence was 71% in standing and 54% in sitting for non-practical students, compared to 39% and 14%, respectively, for practical students. Significant differences were found between groups in sitting (*p* < 0.002) and standing (*p* < 0.001) using TWD, and in CVA measurements (*p* < 0.0001). **Conclusions**: Text neck is more prevalent among non-practical students, while practical students exhibit milder cases. Both sitting and standing assessments are essential for accurate diagnosis and rehabilitation planning. Future studies should analyze the accuracy of various measurement tools to identify the most precise methods for assessing text neck prevalence, taking into consideration both sitting and standing positions.

## 1. Introduction

Text neck, also called forward head posture (FHP), refers to the forward positioning of the head anterior to the plumb line from the sagittal plane [1]. Forward head position to the acromioclavicular joint (AC) joint results in a crossed syndrome, which occurs when the upper cervical vertebrae (C1–C3) are hyperextended, and the lower cervical vertebrae are flexed (C4–C7). The degree of neck displacement anterior to the plumb line determines the severity of text neck, which can be mild, moderate, or severe [2]. According to a qualitative analysis study in India, text neck has the highest percentage prevalence among various postural problems [3].

Many studies have been done in Pakistan, Malaysia, and Indian universities to assess the prevalence of text neck among students, with reported rates ranging between 63% and 73% [4,5,6]. However, these studies did not differentiate between practical and non-practical fields of study. The prevalence of text neck is also high among students younger than university age, with one study reporting a rate of 63% [7]. Another study conducted in Jeddah, Saudi Arabia, using an electronic survey, found a prevalence of approximately 68.1% among university students [8]. While text neck has traditionally been associated with older adults due to age-related mechanical and functional changes, recent trends indicate a rising occurrence among young adults [9,10]. This rising prevalence is linked to multiple factors, such as extended use of electronic devices, physical inactivity, and weight gain.

Text neck has several negative impacts on the body and its functions, including changes in muscle length and performance [10,11]. Also, it affects cervical proprioception, the respiratory system, and induces sleep disorders. Therefore, early assessment and management of text neck, along with the determination of leading factors can lead to a positive impact on the quality of life.

Several methods are used to detect text neck, such as X-ray imaging, 3D motion capture, and posture measurement device (PMD). However, these technologies are not readily available in all clinical settings [12,13]. In addition, there are many objective and valid tools, such as neck posture observation, tragus wall distance (TWD), and craniovertebral angle (CVA) using photography. Several studies have utilized various surface measurement angles to assess text neck severity, including the CVA, head tilt angle, and head position angle [14,15,16].

The most common method for evaluating text neck is the CVA. It is described as one of the methods for determining the alignment of the head relative to the seventh cervical spinous process (C7). The reliability and validity of this angle as a good indicator of text neck severity have been confirmed by numerous studies [15,16,17]. Previous studies have also shown PostureCo and mobile applications to be valid and reliable techniques for assessing CVA [12,17].

While several studies have examined the correlation between electronic device use and text neck [8,18], no study has evaluated text neck prevalence based on academic discipline. There is a lack of research assessing the differences in text neck prevalence between students in practical and non-practical fields. Additionally, no study has investigated the prevalence of text neck among Saudi university students in both sitting and standing positions using valid objective assessment tools beyond questionnaires. We hypothesized that the prevalence of text neck differs significantly between practical and non-practical students, influenced by variations in study routines and ergonomic behaviors.

## 2. Methods

### 2.1. Sample and Subjects

A convenience sample of 82 female students, aged between 18 and 24 years, participated in an analytical observational cross-sectional study (Figure 1). The participants were divided into two groups: practical and non-practical. The practical group consisted of students from the College of Medical Rehabilitation Sciences and the College of Applied Medical Sciences as Physiotherapist and Nutritionist, while the non-practical group included students from the College of Business Administration. All participants were asked about smartphone spending time and the preferred posture in which they spend the most time on the phone. Also, we asked about the study method, duration, and preferred posture to collect comprehensive information about the effect of smartphone spending on neck position. Recruitment was conducted through posters, word-of-mouth communication, and WhatsApp groups.

### 2.2. Ethical Considerations

This study was approved by the Research Ethics Committee of the College of Medical Rehabilitation Sciences, Taibah University (Approval #CMR-PT-2022-02). It adhered to the guidelines of the Saudi National Bioethics Committee, the Declaration of Helsinki, and relevant international standards. Informed consent was obtained, and participant confidentiality was maintained.

### 2.3. Eligibility Criteria

Students with a history of neck or back surgery, prior accidents, cervical spine fractures, scoliosis, severe thoracic kyphosis, torticollis, chronic respiratory conditions, rheumatic diseases, balance impairments, or those using hearing aids were excluded from the study.

Prior to participation, all students received a detailed explanation of the study’s objectives, potential risks, and procedures. They were required to sign an informed consent form and had the right to withdraw at any point. Data collection took place in the Physiotherapy Lab at Taibah University, within a designated research room at the Faculty of Medical Rehabilitation Sciences (female campus), ensuring a controlled environment free from external auditory distractions.

Participants from both practical and non-practical groups underwent an observational postural assessment, categorizing them into four groups: no text neck, mild, moderate, or severe text neck. To objectively measure text neck, two photogrammetric methods—a smartphone-based application and PostureCo—were utilized in both sitting and standing positions. Additionally, the TWD method was applied in the standing position to further assess the severity of text neck.

### 2.4. Assessment Tools

Tools for measurement of height and weight were the regular meter and digital weighing scale (Class body and hydration scale, Model: EF541, Made in Zhongshan Camry Electronic Co., Ltd., located in Zhongshan, China), both of which are valid and reliable [19,20,21].

The TWD measurement for assessing text neck posture is a simple, inexpensive, and a time efficient assessment tool applicable in clinical practice that does not need a specific room design. The TWD measurement has demonstrated high reliability and validity and is very simple in interpretation and could be used for the case follow up [22,23].

Smartphone applications for assessing CVA have emerged as one of the most prevalent methods for evaluating the text neck. Measuring the CVA is the best indicator of the text neck severity, and its reliability and validity have been confirmed in the other previous studies [17,24,25,26]. It is determined by calculating the angle between a horizontal line running through the C7 spinous process and a line extending from the tragus to C7 to assess the text neck. The smartphone application for text neck posture (Physiomaster) is considered a valid and reliable tool to measure the CVA. This method has several advantages as it is reasonably quick, the images and values are obtained easily, and is more accurate and reliable than visual evaluation alone. Because they streamline image acquisition and data analysis while lowering costs, mobile applications have become a viable alternative to photogrammetry systems [27]. Images of each student were captured using a mobile phone (iPhone iOS11).

PostureCo (Trinity, FL, USA) is another method of assessment of CVA by using software application that uses pictures for postural, movement, and body composition analysis [28]. Posture screen mobile analyzes the posture from four views in standing position and only one view from sitting position by Sitscreen. From the anterior view, it analyzes anterior angulation and translation for the head, shoulder, rib cage, hip/pelvic, and knee. From the lateral view, it measures the lateral translation and angulation for the same parts of the anterior view. From the posterior view, it measures the posterior angulation and translation for the head, shoulder, ribcage, hip/pelvic, knee, from T1 to L3, from L3-PSIS midpoint, and ankle pronation or supination. While in the sitting position from the lateral view, it analyzes CVA, head tilt and head neck angle, gaze, thoracic, elbow and wrist angle, thigh angle, and lower leg angle. It also depends on the assessment of CVA by calculating the angle between the tragus and the horizontal line of C7 [28]. The PostureCo application has demonstrated strong rater reliability and preliminary evidence of construct validity (Figure 2A,B) [17].

### 2.5. Procedure

The assessment procedure started by taking history and examination of each student, followed by measuring the weight and height to determine the body mass index (BMI) (weight in kilogram (kg)/height^2^ in meter (m)) [29]. The second step was the observational assessment for text neck from sitting and standing positions [30]. Measurement of the forward head was done by assuming the plumb line which passes through the tragus and shoulder (AC joint). When the tragus is perpendicular to the shoulder, it means no text neck (normal neck posture). Mild text neck is defined as the tragus being forward with the posterior part of the ear lying perpendicular to the shoulder. While moderate text neck is defined as the tragus being forward from the shoulder and the posterior part of the ear being at the edge of the shoulder line. When both the tragus and posterior part of ear are forward from the shoulder, it is considered as severe text neck [17].

The third step involved measuring the distance from the tragus to the wall using a ruler. Each student was asked to stand with the back and pelvic against the wall and feet positioned at the shoulder level. The tragus was determined by placing colored adhesive tape and then using a ruler to measure the distance; how far this land marker was from the wall. If the head touched the wall or is at a distance up to 10–15 cm while naturally standing straight, that means no text neck, and the further the distance, the more text neck [22,23].

The fourth step was to assess CVA by using a smartphone application to capture the sagittal plane profile of the upper body in the sitting and standing positions from the left view for all the students. Before photography, the students were instructed to stand comfortably in natural position with their arms on their sides and to visually focus on a point on the wall directly in front of them while taking an image. Then, the student was asked to sit in a relaxed position and to lean on a backrest and take photo. Land markers were identified first over the tragus and the C7 spinous process by the placed colored adhesive tapes. The C7 spinous process was identified by palpating the lower cervical spine while flexing and extending the cervical spine. The standard distance between the phone and the student was 0.5 m. The phone was set at the student shoulder level. To measure the CVA, the angle between the horizontal line passing through the C7 spinous process and a line extending from the tragus to the C7 spinous process was calculated through a digital photograph on the smartphone application for each student (Figure 3) [17,31,32].

The fifth step of the procedure was PostureCo analysis. After taking the history, each student was assessed for weight by digital weighing scale (kg) and height by regular meter centimeter (cm) and put the results on a sticker with each student. The tape was placed on the floor to indicate where the student should stand for the lateral picture [12]. The tablet (iPad iOS15) was placed on a stand exactly 190 cm away from the student markers in standing and in sitting position 260 cm away at a height of 130 cm to standardize the image angle. Students were instructed to stand at the tape marks in a comfortable position and the feet at the same level of the shoulder and to look at the wall in front of them. While from the sitting position, each student was asked to sit in a comfortable position and simulate writing on the iPad or typing on the laptop (studying position). Students were dressed in medical scrub or clothes showing landmarks with their shoes removed. An image was captured using PostureCo from the standing and sitting position (Figure 4). Then, four specific anatomical landmarks were selected prompted by PostureCo which were the external auditory meatus, the C7 spinous process, the lateral canthus of the eye, and the acromioclavicular joint. Finally, PostureCo calculated CVA using proprietary algorithms [12].

### 2.6. Statistical Methods

#### Sample Size

The sample size calculation for CVA from a standing position, identified as the primary outcome measure, was conducted using the G*Power 3.1 software. The estimation was based on an effect size of 0.85, derived from a pilot study involving 30 students (15 per group). Additional parameters included an alpha level of 0.05, a statistical power of 80%, and a numerator degree of freedom of 1 with two experimental groups [33]. Based on these calculations, the required total sample size for the study was determined to be 46 students, with 23 participants in each group. A total of 110 students were included in the study. However, the final number of students who met the inclusion criteria and were recruited into this study were 82 students, 41 for each group.

The collected data were entered into Microsoft Excel 2019 and analyzed using XLSTATS 2022. *p* < 0.05 was considered statistically significant. The Shapiro-Wilk and Levene tests were used to assess the normality of distributions and the homogeneity of variances. Participants’ characteristics were compared between both groups using a *T* test. Descriptive statistics were used to express the prevalence of text neck for each group and Chi-square analysis was used to detect differences between both groups for the observational analysis of text neck. Because of unequal variance of variables, Welch’s corrected unpaired *T* test (by using One-way ANOVA) was used to detect differences between groups for all the other variables of this paper [34].

## 3. Results

A total of 110 students were screened for eligibility. Of these, 12 did not meet the inclusion criteria, 5 chose not to participate, and 11 were unable to complete the study due to scheduling conflicts with their lectures. The remaining 82 students completed the study. In the practical group, smartphone usage ranged between four to eight hours daily, and the average daily screen time was six hours. While in the non-practical group, smartphone spending time ranges between two to ten hours daily, with an average of nine hours per day on the phone. In studying, the most common method used in the practical group was the iPad, with an average time of five hours daily. However, in the non-practical group, the most common studying method used was both paper and iPad, and they spent around three hours per day studying. Most practical and non-practical groups prefer sitting positions, whether when they are using smartphones or studying. According to normality test by Shapiro-Wilk test, all the variables in both groups were normally distributed. According to Table 1, there was no significant difference between both groups for the demographic data of age, weight, height, and BMI.

### 3.1. Prevalence of Text Neck by Observational Analysis

According to Table 2, the prevalence of text neck in the practical faculties is 80% from the standing position compared to 70% from the sitting position, while in the non-practical faculties, the prevalence of text neck is 91% from the standing position compared to 95.5% from the sitting position.

According to Chi square test, there is a significant difference between both groups for text neck from the sitting position (*p* = 0.002 and Chi square value 15.309) and the percentage for each type of text neck between both groups is demonstrated by (Figure 5), while there is no significant difference between both groups for text neck from the standing position (*p* = 0.351 and Chi square value 3.278), the percentage for each type of text neck between both groups from the standing position is demonstrated by (Figure 6).

### 3.2. Statistical Analysis for Quantitative Measurements of Text Neck

According to Table 3 and Figure 7 and Figure 8, there is a significant difference between both groups for the TWD and Physiomaster measurement of CVA from sitting and standing positions, while there is no significant difference between both groups for CVA PostureCo measurement from sitting or standing position. According to the measurement of TWD, the percentage of text neck is higher in the non-practical group compared to the practical group, based on the criterion that any distance below 15 cm is considered as no text neck [23] (Figure 9). Also, the percentage is higher according to CVA measurements using Physiomaster, based on the criterion that any angle below 48–50 degree is considered text neck (Figure 10) [17,26].

## 4. Discussion

This study assessed the prevalence of text neck in two positions, sitting and standing, due to the differences in the pelvis and thoracic alignment between these positions. In the sitting position, an increase in anterior pelvic tilt leads to greater lumbar lordosis, which contributes to a more pronounced forward neck posture. There is also a difference in the activity of the muscles around the neck, which varies depending on the posture changes between sitting and standing [35]. All these factors emphasize the importance of assessing text neck in both positions, i.e., sitting and standing, rather than relying on a single position to determine the prevalence accurately.

The current study found that text neck was more prevalent among non-practical students compared to practical students, according to observational assessment, TWD, and Physiomaster. Observational assessment showed that text neck was more common in the sitting position among non-practical students, whereas it was more prevalent in the standing position among practical students. While according to Physiomaster, the prevalence was high for both practical and non-practical students from the standing position compared to the sitting position. Moreover, the current study proved that mild and no text neck was dominant among practical students compared to moderate and severe text neck which were dominant among the non-practical students.

This might be attributed to the type of academic discipline which has an impact on the duration of smartphone usage. Both groups in this current study used smartphones; however, according to history, non-practical students had more time available to use smartphones, spending nine hours per day on their phones, with only three hours dedicated to studying. Practical students had an average daily screen time of six hours, with five hours spent studying on iPads. They divided their time between studying on iPads, papers, and practical training in hospitals. The more time spent on laptops, the more the prevalence of text neck. This is agreed with Wiguna et al. [36] and Kang et al. [37].

The prevalence of text neck for both groups in the current study might also be due to faulty posture during lectures, poor ergonomic chair, and lack of knowledge about proper upright posture. This knowledge gap is more pronounced among non-practical students, which might explain why the moderate and severe type of text neck were dominant in the non-practical studying group [4]. Based on the assessment and history consideration in the current study, the non-practical group relied more on laptops and maintained prolonged sitting positions compared to the practical group. This led to more prolonged relaxed sitting and the adoption of a flexed spine posture. On the other hand, practical training in the practical group reduced prolonged sitting time, which may have contributed to a lower prevalence of text neck in the sitting position. However, observational assessment indicated a higher prevalence of text neck in the standing position among practical students, likely because standing was their dominant posture during training. These findings align with Shaghayeghfard B et al. [32] and Vahedi et al. [38], who reported similar trends in posture-related musculoskeletal issues.

CVA measurement using Physiomaster showed that text neck prevalence was higher in the standing position than in the sitting position for both groups, with a greater prevalence in the non-practical group compared to the practical group. In a sitting posture, the lumbar region and neck rotate counterclockwise. When sitting without support, lumbar extension tends to be accompanied by increased flexion in the lower cervical spine. However, in the current study, all students assumed a supported sitting position, which reduced lumbar extension and increased CVA. As a result, CVA was higher in the sitting position, and text neck was more dominant in the standing position. These findings suggest that text neck should be assessed in both sitting and standing positions to determine the most affected posture and develop an effective rehabilitation plan. This conclusion aligns with the findings of Amirdehi et al. [39].

The findings of the present study contradict those of Amirdehi et al. [40], as this study found a difference in the prevalence of FHP between sitting and standing positions, whereas Amirdehi et al. [39] reported no significant difference between the two positions. This discrepancy may be attributed to age-related functional and structural changes, as Amirdehi et al. studied an older population, while the present study focused on young adult students.

The current study contradicts the findings of Cho [41], who reported a low prevalence of FHP (25%), whereas the current study found a significantly higher prevalence in both groups (practical and non-practical), exceeding 60%. This discrepancy may be attributed to differences in the study population, as Cho [41] examined adolescent high school students, while the current study focused on university students. Additionally, methodological differences may have contributed to the variation, as Cho [41] used a survey-based assessment, whereas the present study relied on objective assessment tools. Similarly, the study by Gh et al. [42] reported an FHP prevalence of 24.1% among female participants aged 5 to 20 years, which is significantly lower than the current study’s findings. This discrepancy may also be due to differences in the study population, as Gh et al. [42] focused on younger students, whereas this study examined university students. One possible explanation is that younger participants may have lower overall device usage, leading to a reduced prevalence of FHP in their study. In contrast, university students engage in prolonged screen time for academic and social activities, which likely contributes to the higher prevalence of FHP observed in this study.

The results of the current study did not show any significant difference in text neck prevalence between both groups in either sitting or standing position, as measured by PostureCo. This lack of significance may be due to the assessment position of CVA in the sitting position, as the measurement was taken while participants were using a laptop, which could have lowered CVA in both groups, reducing the likelihood of detecting significant differences. Similarly, the non-significant difference in the standing position may be attributed to the method of taking the picture. The pictures were taken following the guidance of the application software, but the iPad was not always held in a strictly vertical position. Instead, it was often tilted forward. This tilt may have caused inconsistency in the image lengths and quality, affecting the accuracy of the measurements. Additionally, objects positioned further from the camera could have experienced image distortion and miscalculations, further contributing to the absence of significant differences between the two groups.

According to Sarraf et al. and Sheikhoseini et al. [2,43], performing and adhering to corrective exercises may help reduce the prevalence of text neck and manage its impact and progression effectively. Corrective exercises that may help reduce text neck severity and improve posture include deep cervical flexor muscles exercise (chin tuck), isometric strengthening of the cervical flexor muscles, and isotonic strengthening of the scapulothoracic muscles [43]. Additionally, manual therapy and stabilizing exercises have shown significant improvements in pain, function, and forward head posture [44]. Fathollahnejad et al. [44], emphasized the importance of maintaining neutral posture during daily activities and exercises. In their study, participants were guided using mirrors placed in front and at the sides to ensure proper posture. Furthermore, Titcomb et al. [45], suggested that following proper postural instructions while using mobile phones or computers may help reduce or prevent text neck among students. Postural education for smartphone and computer users includes keeping the head aligned vertically with the spine while sitting or standing, supporting the elbows and raising the mobile device to minimize forward head tilt. For computer users, it involves maintaining a 90-degree elbow flexion, supporting the lower back while sitting, and avoiding prolonged static postures.

### Limitations and Recommendations

This study had several limitations. First, all participants were female, which may limit the generalizability of the findings to a broader population. Additionally, the use of convenience sampling may have affected the representativeness of the sample. Another limitation was the reliance on observational assessment, which, despite the use of predefined criteria, remains inherently subjective and susceptible to inter-rater variability; however, inter-rater reliability was not assessed in this study. Furthermore, research on CVA measurement using PostureCo analysis remains limited, which may affect the interpretation and reliability of the results. Given these limitations, future studies should aim to explore text neck prevalence across diverse practical fields, such as dentistry and computer science, and validate PostureCo as a reliable tool for CVA measurement to enhance assessment accuracy. Additionally, studies should explore text neck prevalence in diverse populations, including males and different age groups, to enhance generalizability.

## 5. Conclusions

This study revealed a high prevalence of text neck among both non-practical and practical students at Taibah University, with a significantly higher prevalence among non-practical students. Additionally, CVA measurements indicated a decrease in CVA values, reflecting a higher prevalence of text neck in both sitting and standing positions. Physiomaster CVA measurements showed a greater prevalence of text neck in the standing position for both groups, while observational analysis revealed a higher prevalence in the sitting position, particularly among non-practical students. These findings suggest that both sitting and standing postures should be considered when assessing text neck to determine the most affected position and develop targeted rehabilitation strategies. Future research should focus on evaluating the accuracy of different measurement tools to establish the most reliable methods for assessing text neck prevalence across various postures.

## Figures and Tables

**Figure 1 healthcare-13-00651-f001:**
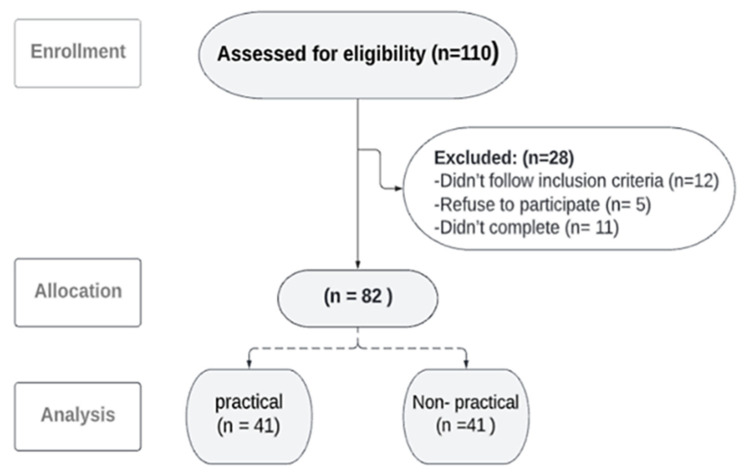
Flow chart of students through study.

**Figure 2 healthcare-13-00651-f002:**
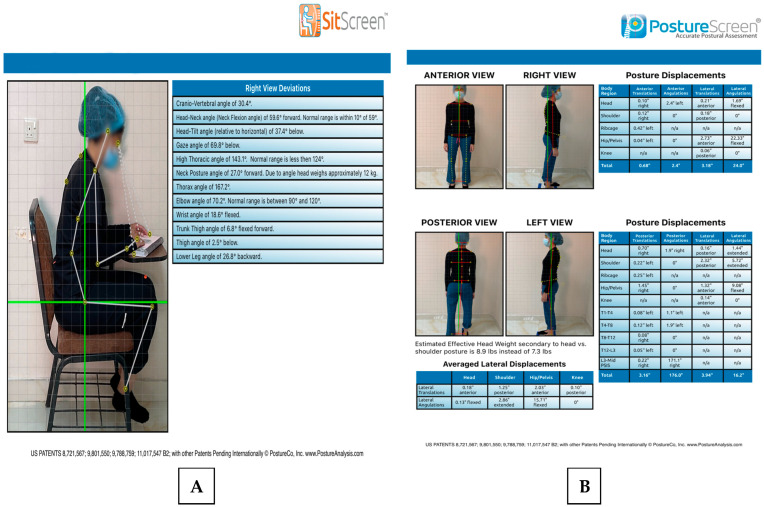
(**A**) Sitscreen and (**B**) Posture screen mobile 4-view.

**Figure 3 healthcare-13-00651-f003:**
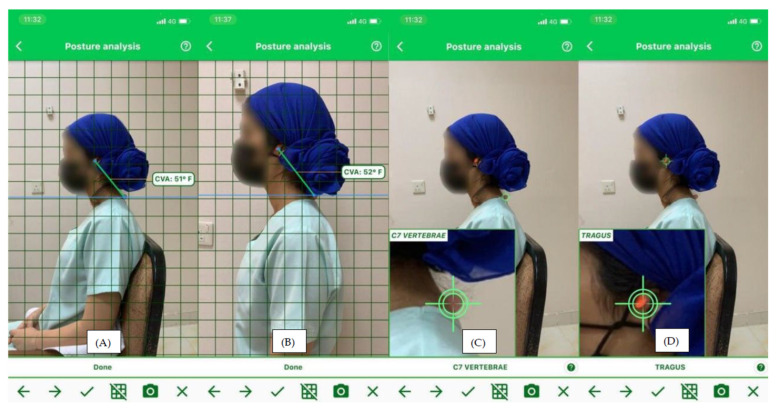
(**A**) Locating tragus point, (**B**) Locating C7 point, (**C**) The angle formed by a horizontal line passing through the C7 spinous process and a line extending from the tragus of the ear to the C7 spinous process is used to calculate CVA and (**D**) CVA from standing position.

**Figure 4 healthcare-13-00651-f004:**
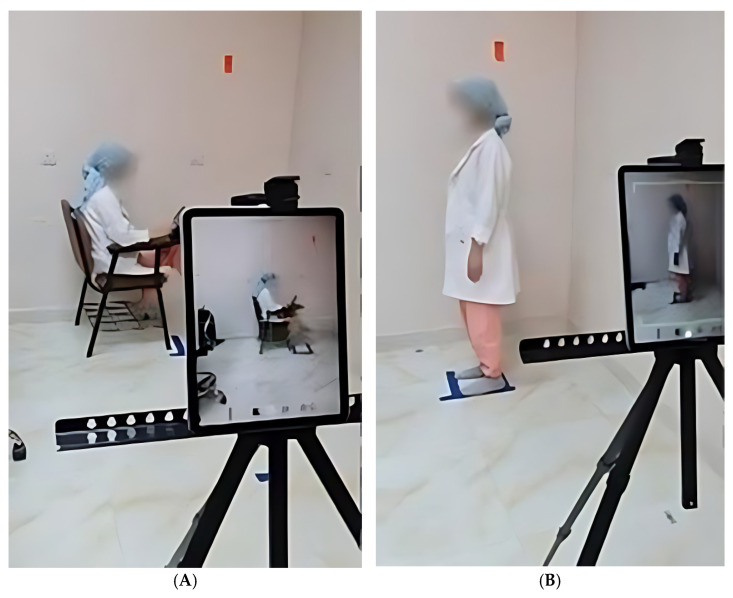
Sitting (**A**) and standing position (**B**) PostureCo capture.

**Figure 5 healthcare-13-00651-f005:**
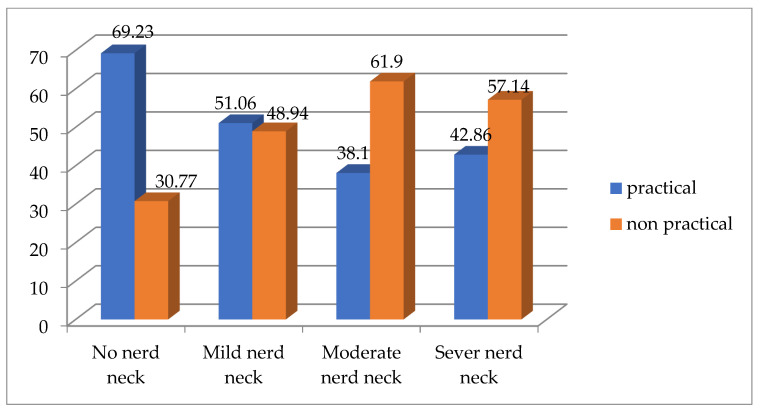
Comparing the percentage of each type of text neck between both groups from sitting position.

**Figure 6 healthcare-13-00651-f006:**
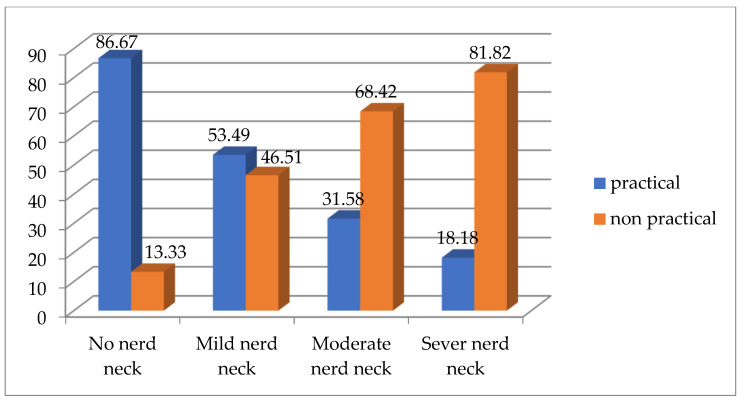
Comparing the percentage of each type of text neck between both groups from standing position.

**Figure 7 healthcare-13-00651-f007:**
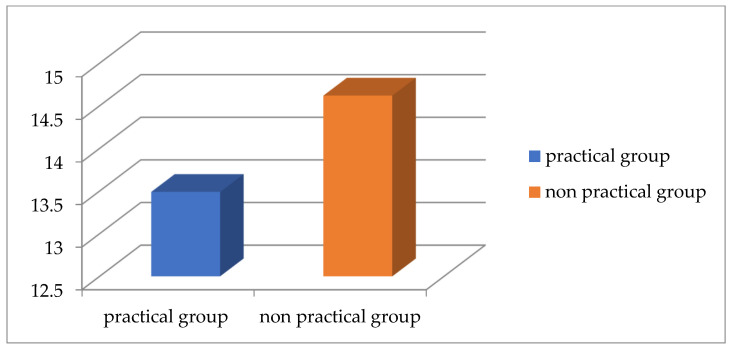
The mean values of TWD for both groups.

**Figure 8 healthcare-13-00651-f008:**
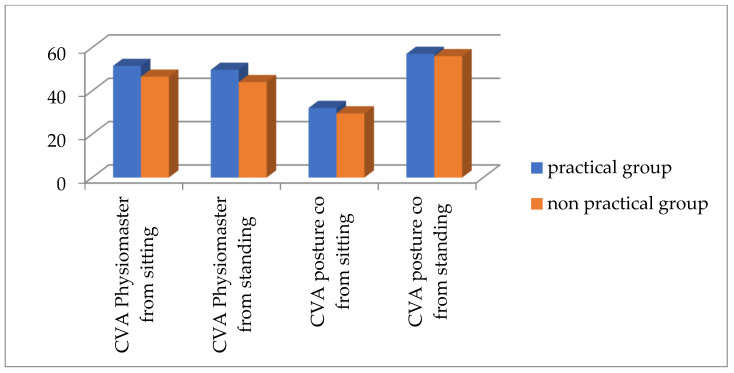
The mean values of CVA for both groups by Physiomaster and PostureCo.

**Figure 9 healthcare-13-00651-f009:**
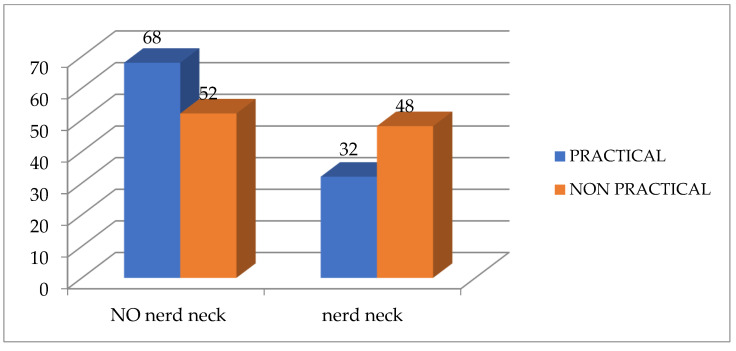
Percentage of text neck and no text neck between both groups according to TWD measurement.

**Figure 10 healthcare-13-00651-f010:**
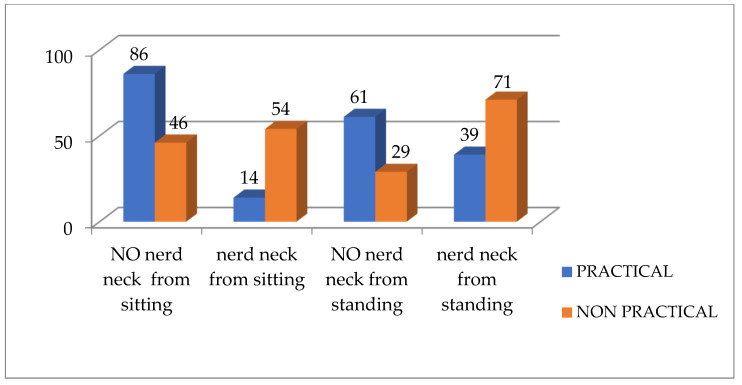
Percentage of text neck and no text neck between both groups according to CVA measurement using Physiomaster.

**Table 1 healthcare-13-00651-t001:** The mean values of demographic data for both groups.

Variables	Practical Group		Non-Practical Group		*T* Test	*p*-Value
	Mean Values	SD	Mean Values	SD		
Age (yo)	21.1	1.1	20.9	1.4	1.99	0.49
Weight (kg)	53.88	11.9	49.66	8.9	1.99	0.07
Height (cm)	157.85	7.1	153.98	13.7	1.99	0.11
BMI (kg/h^2^)	21.61	4.7	22.77	16.7	1.99	0.67

yo: years old; kg: kilogram; cm: centimeter; h: height.

**Table 2 healthcare-13-00651-t002:** The percentage of each type of text neck for both groups from sitting and standing positions.

Text Neck Percentage	Normal Neck	Mild Text Neck	Moderate Text Neck	Severe Text Neck
Standing practical group	20%	55%	18%	7.00%
Standing non-practical group	9%	52.00%	30.00%	9%
Sitting practical group	30%	52%	14%	4.00%
Sitting non-practical group	4.50%	45.50%	29.50%	20.50%

**Table 3 healthcare-13-00651-t003:** Mean values of quantitative measurement of text neck from sitting and standing positions for both groups.

Variables	Practical Group		Non-Practical Group		F Test	*p* Value
	Mean Values	SD	Mean Values	SD		
The tragus wall distance	13.49	1.7	14.62	1.4	11.36	0.001 *
CVA Physiomaster from sitting position	51.53	5.1	46.57	5.9	18.04	0.0001 *
CVA Physiomaster from standing position	49.74	7.01	44.06	5.3	18.11	0.0001 *
CVA PostureCo from sitting position	32.10	9.8	29.59	11.1	1.36	0.247
CVA PostureCo from standing position	57.08	4.6	56.03	4.6	1.31	0.255

* = significant.

## Data Availability

The original contributions presented in this study are included in the article. Further inquiries can be directed to the corresponding author.

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
