# Peer review of "Prevalence of Text Neck Between Different Female Students at Taibah University, Saudi Arabia; Cross Section Design"

_healthcare, 2025, doi:10.3390/healthcare13060651_

Round 1

Reviewer 1 Report

Comments and Suggestions for Authors

Dear Authors,

First, I would like to commend you for your efforts in conducting this important study. The manuscript provides a well-designed investigation into the prevalence of text neck among female university students in Saudi Arabia. While the study offers valuable insights, there are several areas that require clarification and revision before it can be considered for publication. Please address the specific comments outlined below:

Abstract:

  • Please add the study objective just before the "Methods" section.
  • Line 14: Kindly clarify the distinction between the “practical” and “non-practical” groups.
  • The results section (line 17 onwards) should be rephrased. Consider presenting the percentage values directly within the statements for clarity.
  • Line 25: The conclusion section needs to be more robust, with clearer recommendations for future actions based on the findings.

Introduction:

  • The introduction requires further development to enhance the flow and cohesiveness. Some paragraphs deviate from the typical structure, so I suggest reorganizing them for better clarity.
  • Please include a regional perspective, as the focus of your study is specific to Saudi Arabia.
  • Line 37: When referring to "many studies," please provide more specificity. It would be helpful to mention key studies or briefly summarize the findings.
  • Line 57: The phrase “several studies” is accompanied by only one citation. Please review similar instances in the text and ensure that the references align properly.
  • Lines 66-74: This section lacks clarity. Please reformulate it to more clearly define the research problem, identify gaps, and highlight the originality of your study. Avoid excessive use of "no study..." statements in succession. Additionally, it would be helpful to include the study's objectives prior to the hypotheses.

Methods:

  • Line 79: The term "practical" needs further clarification, as it may not be clear to all readers. Please provide a more specific definition.
  • Line 106: Please double-check the placement of the "2.4. Assessment tools" subsection.
  • Line 213: The placement of "low chart of students through study" is unclear. It may be more appropriate to integrate this within the "Sample and Subjects" section.

Results:

  • I have no major concerns with the results section. However, the manuscript includes a large number of figures. I recommend consolidating some of these into single figures or removing any that are not essential for conveying the key findings.

Discussion:

  • The discussion section would benefit from a deeper engagement with the existing literature. Currently, the study's findings are not adequately compared or contrasted with previous research. The number of references used in the discussion is relatively low.
  • Therefore, I strongly recommend rewriting the discussion to better contextualize the study's findings within the broader field.
  • Lastly, the "Conclusions" section could be strengthened by incorporating more concrete and actionable recommendations based on the study's results.
Comments on the Quality of English Language

N/A

Author Response

Dear reviewer, 

Thank you for your valuable comments. Please see the attached documents. 

Best Regards

Reviewer 2 Report

Comments and Suggestions for Authors

The paper by Mashabi et al is a cross-sectional study investigating the prevalence of text neck among female students at Taibah University in Saudi Arabia, comparing practical and non-practical student groups.

The study included 82 female students aged 18-24 years, divided into practical (students from the College of Medical Rehabilitation Sciences and the College of Applied Medical Sciences) and non-practical groups (students from the College of Business Administration). Text neck was evaluated through observational postural assessment, TWD measurement, and CVA measurement using the Physiomaster app and PostureCo software in both sitting and standing positions.

Authors conclude that text neck is more prevalent in non-practical students, with practical students exhibiting milder forms.

I compliment the authors. The topic is relevant given the increasing use of electronic devices and the potential impact on posture. Research on text neck prevalence within specific student populations is valuable.

Possible are of improvement:

  • The use of convenience sampling may introduce selection bias, as the participants may not be representative of the entire student population. This should be acknowledged in the limits.
  • While the observational assessment provides context, it is inherently subjective and prone to inter-rater variability, even with defined criteria. The paper does not mention inter-rater reliability for this assessment. This should be acknowledged in the limits.
  • The paper mentions the relationship between electronic device usage and text neck, but does not include data on the average daily screen time of participants, or the type of devices they commonly use. This information could have helped to reinforce the link between tech use and posture.
  • Authors could strengthen the Discussion. Providing a more in-depth discussion of the implications of the findings for clinical practice and public health would enhance the paper. How can the results inform interventions to prevent or manage text neck in university students? Do authors think that education could help in reducing text neck prevalence? You could discuss this, comparing findings from a recently published paper on a similar topic (https://doi.org/10.1007/s00068-025-02788-9).
  • I suggest authors revise English: some sentences are difficult to follow and understand, even if correct. Moreover, words repetitions are present, reducing reading fluidity.
    • Examples:
    • lines 37-40 “Many studies have been done to assess the prevalence of text neck among students as a general without differentiation between practical and non-practical studying type and most of the studies showed prevalence ranging between 63% to 73%, according to studies which were done in the Pakistan and Indian universities, respectively”
    • lines 66-68: “As per the research carried out in this report, there are many studies that were done to assess the correlation between using electronic devices and text neck, but no study was done to assess text neck prevalence according to the type of studying.” The word study or studying is repeated several times.
    • Line 144: “It started by taking full history…” it is not clear what “it” refers to.
    • Line 214: “Data were entered into Microsoft Excel 2019 and analyzed XLSTATS 2022.”
    • Lines 292-298: “As both groups in this current study used smart phone but as the load of studies was more in the practical group; according to the history of both groups, the non-practical students have more time available to use smart phones. Also, most non- practical students depended mainly on laptop while practical students divided their time between studying on laptop, paper and practical training in hospitals and the more the time spent on laptop the more the prevalence of text neck. This is agreed with Wiguna et al. [34] and Kang et al. [35]”
  • Authors talk about a previous “pilot study involving 30 students (15 per group)” – line 205. The study is however not cited.
Comments on the Quality of English Language

see my comments for detail on how to improve language

Author Response

(The authors gave the same response as above.)

Round 2

Reviewer 1 Report

Comments and Suggestions for Authors

Dear authors,

Thank you for revising the manuscript and for your responses. The revisions made have improved the clarity and overall structure of the manuscript.

Author Response

Dear Reviewer, 

Thank you for your comments and feedback.

Best Regards,

Reviewer 2 Report

Comments and Suggestions for Authors

I would like to thank the authors for the great effort spent in answering my comment. I see some improvement, but some adjustment still needs to be made.

  • In my previous report, I asked about the time participants spent on smartphones. Authors added this information in the discussion. However, presenting this data in that location in my opinion lacks clear methodology. Authors should express in the methods section the way they assessed this “time spending”, then present the average data (adding ranges) in the results and lastly discuss these results.
  • While discussion has been improved a little, the simple addition of two references is not enough in my opinion (lines 375-385). I also suggest authors to revise English: the sentence “According to Sarraf et al. That to manage and decrease the prevalence of compline of neck text by doing a corrective exercise [42].” is without any meaning. Perhaps the “dot” is simply misplaced, however even considering this option the sentence needs to be revised.
  • In the same location (lines 375-385) authors report to have “compared our findings with the recent study (https://doi.org/10.1007/s00068-025-02788-9)”. However, both the comparison and the reference (i.e. Migliorini, F., Maffulli, N., Schäfer, L. et al. Impact of education in patients undergoing physiotherapy for lower back pain: a level I systematic review and meta-analysis. Eur J Trauma Emerg Surg 51, 113 (2025). https://doi.org/10.1007/s00068-025-02788-9) are not present there or in other sections. Please check it.
  • An additional general English revision is needed.
Comments on the Quality of English Language

please, see my comments 

Author Response

Dear Reviewer, 

Thank you for your comments and feedback. Please see the attached file. 

Best Regards,
